# Hetero-Bimetallic Ferrocene-Containing Zinc(II)-Terpyridyl-Based Metallomesogen: Structural and Electrochemical Characterization

**DOI:** 10.3390/ma16051946

**Published:** 2023-02-27

**Authors:** Evelyn Popa, Adelina A. Andelescu, Sorina Ilies (b. Motoc), Alexandru Visan, Carmen Cretu, Francesca Scarpelli, Alessandra Crispini, Florica Manea, Elisabeta I. Szerb

**Affiliations:** 1“Coriolan Drăgulescu” Institute of Chemistry, Romanian Academy, 24 Mihai Viteazu Bvd., 300223 Timisoara, Romania; 2MAT-INLAB (Laboratorio di Materiali Molecolari Inorganici), Dipartimento di Chimica e Tecnologie Chimiche, Universitá della Calabria, Arcavacata, 87036 Rende, Italy; 3Department of Applied Chemistry and Engineering of Inorganic Compounds and Environment, Politehnica University of Timisoara, Bvd. Vasile Parvan No. 6, 300223 Timisoara, Romania

**Keywords:** Zn(II)/ferrocene metallomesogen, hetero-bimetallic coordination complex, PXRD studies, cyclic voltammetry, electrochemistry

## Abstract

The synthesis, as well as the mesomorphic and electrochemical properties, of a hetero-bimetallic coordination complex able to self-assemble into a columnar liquid crystalline phase is reported herein. The mesomorphic properties were investigated by polarized optical microscopy (POM), differential scanning calorimetry (DSC) and Powder X-ray diffraction (PXRD) analysis. Electrochemical properties were explored by cyclic voltammetry (CV), relating the hetero-bimetallic complex behaviour to previously reported analogous monometallic Zn(II) compounds. The obtained results highlight how the presence of the second metal centre and the supramolecular arrangement in the condensed state pilot the function and properties of the new hetero-bimetallic **Zn/Fe** coordination complex.

## 1. Introduction

Functional homogeneous materials obtain single properties derived from functional moieties synthetically inserted into a molecular structure in a unique result that is not only the sum of these properties; often, new and synergistic properties are obtained, encouraging further research in this direction. However, the design of functional molecular structures to obtain performances for a certain application is still serendipitous; hence, fundamental research linking molecular structure-properties for structural complex molecules is compulsory. Metallomesogens (MMs) are functional materials combining the order and anisotropy properties of liquid crystals with the properties derived from a metal centre [1,2], resulting in polarized emission [3,4,5,6,7,8], peculiar magnetic properties [9,10,11,12,13,14,15], enhanced thermal and electrical conductivities [16,17,18,19], charge carrier mobilities [7,20,21,22,23], etc.

Previously, we showed that Zn(II) metallomesogens are promising candidates in electrochemical sensing, as they are able to generate hierarchically ordered metal oxide (MOx) nanoelectrode arrays in situ after electrochemical treatment [24]; moreover, the investigation of three electrode compositions obtained by varying the ratio between carbon nanotubes and Zn(II) metallomesogen showed the importance of both the weight ratio and supramolecular arrangements in the liquid crystalline state [25]. 

The insertion of a second metal centre into the molecular structure of MMs yields binuclear homo-metallic or hetero-bimetallic liquid crystals with an increasing structural and supramolecular complexity and further possible improvements or induction of synergistic properties. Some examples of binuclear metallomesogens found in the literature are synthetic challenges with investigation of their supramolecular “soft” assemblies [26,27,28,29,30,31,32]; in one case, the magnetic properties were determined [33]. However, due to the high transition temperatures typical for these hetero-bimetallic metallomesogens, the enthusiasm for synthesising such systems has diminished. Herein, we wish to recall attention to bimetallic MMs and to explore their electrochemical properties for sensing applications. 

Against this background, a new coordination complex, namely **Zn/Fe**, the chemical structure of which is presented in Figure 1, was synthetized and structurally characterized.

The supramolecular architecture of complex **Zn/Fe** in the mesophase and its electrochemical properties were investigated and is presented in discussion with a previously reported analogue, complex **Zn** (Figure 1) [3], in an attempt to relate the presence of an additional metal center on the molecular structure - supramolecular structure – electrochemical property relationship. The electrochemical properties were determined using CNT paste working electrodes modified by a simple film-casting method with **Zn/Fe** and **Zn** coordination complexes (Figure 1) and a ferrocene-containing precursor (**S4**—Figure 1) named **Zn/Fe_CNT**, **Zn_CNT** and **Fc_CNT** paste electrodes. Their electrochemical behaviours were comparatively studied by cyclic voltammetry (CV) in 0.1 M NaOH supporting electrolyte.

## 2. Results

### 2.1. Synthesis 

The synthesis of complex **Zn/Fe** is presented in Figure 1. Compounds **S1** and **S2** were synthesised by adapting a literature procedure [34], compound **S3** was obtained as a byproduct of the Williamson etherification reaction between ethyl gallate and *n*-bromododecane and the ligand 4-(4-N,N-diethylbenzenamine)-2,6-di(pyridine-2-yl)pyridine (**S5**) was synthesized according to a literature procedure [35]. 

The synthesis of the final **Zn/Fe** coordination complex was carried out using a modified synthetic strategy with respect to that reported for the analogous **Zn** complex [3], which required the use of a silver salt of the benzoate precursor ligand (**S4**) and the formation of a Zn(II) dichloro derivative of the chelating ligand (**S5**). However, the presence of silver ions caused oxidation of the ferrocene units. The new complexation reaction was carried out by a simplified method adapted from a literature procedure [36]. In particular, the sodium salt of precursor **S4** was formed and then reacted with ZnCl_2_. The resulting intermediate was used directly in the next step without purification. After a chelation reaction with ligand **S5**, the pure product was obtained by repeated recrystallisation, with improved yields. While the total yield of the complexation reaction that used the silver salt benzoate was 55% [3], herein, we obtained complex **Zn/Fe** in 70% yield.

Complex **Zn/Fe** was structurally characterized by FT-IR and ^1^H and ^13^C NMR spectroscopy, while its purity was determined by elemental analysis (see Appendix A: Appendix A). The IR spectrum contains the vibrational bands related to the two ligands and the ferrocene unit. In particular, the spectrum of complex **Zn/Fe** is almost superimposable on the spectrum of complex **Zn**, additionally containing the vibrational bands of a ferrocene unit centred at 1002, 924 and 486 cm^−1^ (ν_Fc_) [37]. The similar spectra and the separation of the stretching vibrations of COO^−^ group (Δ) of 259 cm^−1^ (see Appendix A: Appendix A and Appendix A) suggests the same molecular structure for complex **Zn/Fe** as that for complex **Zn**: a neutral species with the metal centre pentacoordinated by a chelating tridentate terpyridine-based ligand and two monoanionic benzoate derivatives. Moreover, the coordination environment around the metal centre and the proposed molecular structure of complex **Zn/Fe** is supported by similar structures reported in the literature [38,39,40,41].

The structure and purity of the final **Zn/Fe** complex was confirmed by elemental analysis, as well as ^1^H , ^13^C NMR and AAS (see Appendix A: Appendix A and Appendix A). The thermal stability and presence of crystallisation solvent were determined by thermogravimetric analysis (TGA). A loss of one molecule of water was observed at around 50 °C, while the complex showed good thermal stability [42]; the weight loss from degradation became significant above ca. 300 °C and reached 5% at T_5%_ = 324 °C (see Appendix A Appendix A). The majority of the organic part decomposed between 300 and 500 °C, leaving residual zinc and iron oxides accounting for 8.17%.

### 2.2. Mesomorphism

The mesomorphic properties of complex **Zn/Fe** were determined by polarized optical microscopy (POM), differential scanning calorimetry (DSC) and powder X-ray diffraction (PXRD) studies. 

#### 2.2.1. POM and DSC Studies

During the first heating of the pristine complex, a first broad endothermic event spreading over several tens of degrees Celsius with a large enthalpy was detected by DSC (Figure 2b), while the complex transited into a birefringent fluid phase. Upon further heating, the birefringence persisted until the sample fully cleared at around 150 °C. Upon cooling, the complex arranged into a hexagonal columnar mesophase as can be derived from the mix mosaic texture (Figure 2a) and the presence of homeotropic zones. Upon further cooling, no sign of crystallisation could be detected either by POM or by DSC, while upon further heating and cooling, only one transition was observed by DSC (Figure 2b). The optical texture of the mesophase was preserved upon cooling, indicating the formation of anisotropic liquid crystalline glasses. The complexes have a high thermal stability, as demonstrated by repeated heating–cooling cycles.

#### 2.2.2. PXRD Studies

The mesophase of complex **Zn/Fe** was investigated through PXRD analysis. The diffraction pattern of the Zn(II) complex, recorded at 85 °C upon cooling from the isotropic liquid (Figure 3a), presents an intense reflection at 2θ = 3.3° (d = 26.8 Å, Table 1) and other less intense reflections in the small–middle-angle region. This pattern can be indexed on the basis of a hexagonal columnar system (Col_hex_), with the first reflection assigned to the (10) interplanar distance. In addition, the semibroad diffraction peak (h_0_) in the wide-angle range precisely at 2θ = 26.2° (d = 3.4 Å) can be assigned to intracolumnar π-π stacking. It is reasonable to assume that, analogously to the recently reported parent **Zn** complex [3], the columnar self-assembly of molecules is driven by the instauration of π-π stacking between the terpyridine (tpy) cores. Moreover, the tpy groups must be rotated 120° ca. relative to each other in order to uniformly distribute the aliphatic chains around the aromatic cores [3]. It is worth noting that the parent **Zn** complex is characterized by a 3D hexagonal mesophase, with a high degree of intracolumnar order, whereas the **Zn/Fe** compound generates a less ordered 2D mesophase. This observation can be explained by the fact that in complex **Zn/Fe**, the ferrocene moiety, which is incorporated in two alkyl chains, probably interferes with the periodic segmentation of columns responsible for the three-dimensional order observed in the mesophase of complex **Zn** [3]. 

Finally, once generated upon cooling from the isotropic state, the mesophase of complex **Zn/Fe** remains frozen at room temperature (Figure 3b), with minimal *d* and cell parameter changes (Table 1).

It is worth recalling that the analogues structural Zn(II) coordination complex based on a non-substituted terpyridine (lacking the apical N,N-diethylbenzenamine group) reduces the symmetry of the complex, and despite the pentacoordination around the metal centre and the bulky, voluminous structure, the resulting complex arranges into smectic-type liquid crystalline phases [24].

### 2.3. Electrochemistry

A CNT paste working electrode was modified by a simple film-casting method with **Zn/Fe** metallomesogen to obtain the **Zn/Fe_CNT** paste electrode, which was characterized in comparison with the paste electrodes based on the **Zn** coordination complex and the ferrocene-containing precursor (**S4**) (**Zn_CNT** and **Fc_CNT**, respectively) obtained under similar conditions.

The electrochemical behaviour of **Zn/Fe_CNT** was studied by cyclic voltammetry (CV) within the potential range of −1.5 V/SCE to +1 V/SCE considering the manifestation of both ferrocene and ferrocenium redox systems [44,45,46]. The comparative voltammograms recorded in 0.1 M NaOH supporting the electrolyte on **Zn/Fe_CNT** in comparison with **Zn_CNT** are shown in Figure 4a, while a comparison with the ferrocene precursor (**Fc_CNT**) is presented in Figure 4b. The voltammetric behaviour of the liquid crystalline zinc coordination complex organised in a smectic mesophase previously reported by our group in reference [24] was also considered to evaluate an eventual influence of the type and symmetry of the mesophase on the electrochemical response of the paste electrode.

To study the comparative electrochemical behaviours, each material was used as a modifier for the CNT paste electrode by film-casting method. The significant differences between the current values are due to the varying degree of the CNT paste electrode surface coverage ***(****Γ*) (Table 2), as determined for each material based on the relation between peak current (CV curves) and the scan rate [47] using equation (1):(1)$$I_{p}=n^{2}\times F^{2}\times v\times A\times \Gamma /4R\times T$$where *n* is the number of exchanged electrons, *F* is the Faraday, *A* is the area of the electrode surface and *Γ* is the surface coverage.

The different degrees of the CNT paste electrode surface coverages achieved by film-casting method with **Zn/Fe**, **Zn** coordination complexes or the **S4** precursor are due to the different material consistencies and sorption affinities for CNT.

The shape of CVs recorded for the **Zn_CNT** paste electrode was very similar that recorded for the liquid crystalline Zn(II) coordination complex organised in the previously reported smectic mesophase [24], except the potential of the cathodic peak characteristic of metallic zinc and zinc oxide formation, which is less negative for the **Zn_CNT** paste electrode. This may be related to the different supramolecular arrangement in the mesophase (column vs. layers).

The anodic branch of the CV shape recorded on the **Fc_CNT** electrode is in accordance with the reported literature regarding ferrocene/ferrocenium redox couples based on reaction (2) [48]:(2)$$Cp_{2}Fe↔Cp_{2}Fe^{+}+e^{-}$$ within the cathodic branch, one oxidation peak occurred at about −0.66 V vs. SCE and the corresponding reduction peak at about −1.00 V vs. SCE, which corresponds to the Fe/Fe(II) redox couple [49,50,51,52]. This couple is not evidenced for **Zn/Fe_CNT** because it is probably overlayed with zinc reduction and stripping couples.

All anodic peaks corresponding to zinc stripping and zinc oxide dissolution and all cathodic peaks corresponding to zinc reduction and zinc oxide formation under oxygen reduction reaction conditions are manifested for **Zn/Fe_CNT** and **Zn_CNT** paste electrodes, which assure the presence of zinc redox systems. However, a higher magnitude of current is manifested in the former due to the overlay of both zinc and ferrocene redox systems. The ferrocene/ferrocenium redox couple is manifested in the **Zn/Fe_CNT** paste electrode, with the reductive back peak less defined than the oxidative forward peak. This may be due to the incorporation of ferrocenium within the Zn(II) coordination complex structure, which stabilises it sufficiently such that its reduction becomes thermodynamically unfavourable [48].

To accurately study the effect of ferrocene grafted into the molecular structure of Zn(II) metallomesogen and the contribution of the different supramolecular arrangements (a 2D columnar hexagonal phase in **Zn/Fe** and a 3D columnar hexagonal phase in **Zn** metallomesogens) to the electrochemical features, the scan rate effect on the shapes of CVs recorded on each of the **Zn/Fe_ CNT**, **Zn_CNT** and **Fe_CNT** paste electrodes was investigated. The series of CVs recorded at scan rates ranging from 0.01 V∙s^−1^ to 0.20 V∙s^−1^ on all electrodes is presented in Figure 5a–c.

For each modified electrode, the anodic and cathodic peaks are marked and presented in Table 3 for comparison. All anodic and cathodic peak currents increased linearly with the square root of the scan rate (Appendix A) because all anodic and cathodic processes described in Table 3 are diffusion-controlled.

It can be noticed that the **Zn/Fe_CNT** paste electrode based on the 2D columnar hexagonal metallomesogen exhibited electrochemical behaviour based on the combination of the **Zn_ CNT** paste electrode (3D columnar hexagonal metallomesogen based on a similar molecular structure Zn(II) coordination complex lacking ferrocene units) and the **Fc_CNT** paste electrode (precursor containing ferrocene units). In the cathodic branch, the electrochemical behaviour of the **Zn** complex is predominant (no Fe/Fe(II) redox couple is manifested), while in the anodic branch, the ferrocene/ferrocenium redox couple is better manifested. An anodic peak is evidenced at a potential value of about +0.290 V vs. SCE (AI) before a ferrocene oxidation peak (AII) recorded at about +0.600, which can be attributed to zinc oxide dissolution with the formation of zincate. This peak is clearer at the higher scan rate, evidencing fast kinetics of the zinc oxidation dissolution process.

Considering the integration of ferrocene units into the molecular structure of the Zn(II) coordination complex, the redox couple of ferrocene/ferrocenium was comparatively analysed in the **Zn/Fe_CNT** paste electrode and the **Fc_ CNT** paste electrode. The details of various current and potential parameters determined based on Figure 5b,c are presented in Table 4 for the **Fc_CNT** paste electrode and in Table 5 for the **Zn/Fe_CNT** paste electrode.

As the scan rate was increased, the potential value of the anodic peak shifted to more positive values, while the cathodic peak values became more negative. Smaller differences between the anodic and cathodic potential values (ΔE_p_) were obtained at a low scan rate than at a fast scan rate (Table 4 and Table 5) for both electrodes. A larger difference was obtained for the **Zn/Fe_CNT** paste electrode (0.310 V) than the **Fc_CNT** paste electrode (0.020 V) at a scan rate of v = 0.01 Vꞏs^−1^. When the scan rate was increased by 20 times, an almost 1.45-fold increase in ΔE_p_ was observed for the **Zn/Fe_CNT** paste electrode in comparison with a 10-fold increase for the **Fc_CNT** paste electrode. The high ΔE_p_ value indicates a quasireversible charge–transfer process. When the scan rates increased, the changes in the ΔE_p_ values provided information about the partial control of the charge–transfer step relative to the diffusion step. Theoretically, there no change occurred in ΔE_p_ with the scan rates when the charge–transfer was fast and completely reversible [44]. Additionally, the ratio between the anodic and cathodic peak currents should be close to the theoretical value of 1 for all the scan rates used for a reversible system. Taking these factors into considerations our findings were corroborated; the Fc/Fc+ couple was closer to the ideal reversible system in the **Fc_CNT** paste electrode considering the ΔE_p_ values and the ratio between the anodic and cathodic peak currents. However, smaller changes of the ΔE_p_ values with scan rates were observed for the **Zn/Fe_CNT** paste electrode, indicating a faster charge–transfer rate. The occurrence of electron transfer to and from the redox centres of the ferrocene is also indicated by the linearity of the anodic peak current vs. the square root of the scan rate for both electrodes (Appendix A).

## 3. Materials and Methods

### 3.1. Synthesis

All commercially available starting materials and solvents were used as received without further purification. Ferrocene, anhydrous AlCl_3_, NaBH_4_, anhydrous dichloromethane and tetrahydrofuran were purchased from Sigma Aldrich, while hexane and ethyl acetate were purchased from Carlo Erba; HPLC-grade dichloromethane was purchased from Honeywell. A Bruker Avance III HD—500 MHz spectrometer was used to record ^1^H and ^13^C NMR experiments in CDCl_3_ or CD_2_Cl_2_. A Flash 2000 microanalyser from Thermo Fisher Scientific was used to perform elemental analyses (CHN), while the percentage of Zn(II) was determined using a SensAA flame atomic absorption spectrometer (GBC Scientific Equipment, Braeside, Australia). The instrument was equipped with a zinc hollow cathode lamp (detection limit: 0.4–1.5 mg/L, integration time 3 s). The flame used was an air–acetylene mixture. Two determinations were made, and the average absorbance value was used.

### 3.2. Optical and Thermal Studies

An Olympus BX53M polarizing microscope (POM) equipped with a Linkam hot stage and an Olympus UC90 camera was used to observe the optical mesophase textures of complex Zn/Fe. Thermal decomposition was carried out using a TGA/SDTA 851-LF 1100 Mettler Toledo thermogravimetric analyser, with the experiments conducted in a nitrogen atmosphere in the temperature range of 25–800 °C with a heating rate of 10 °C min^−1^. Enthalpies and transition temperatures were recorded using a Q1000 apparatus from TA Instruments. The apparatus was calibrated with indium; three heating/cooling cycles were performed for each sample, with a heating and cooling rate of 10 °C/min.

### 3.3. Powder X-ray Diffraction (PXRD) Analysis

A Bruker D2-Phaser equipped with Cu Kα radiation (λ = 1.5418 Å) and a Lynxeye detector at 30 kV and 10 mA with a step size of 0.01° (2θ) were used to record the PXRD patterns of the **Zn/Fe** mesophase. The sample was heated and cooled at a rate of 5 °C min^−1^ using a Sil’tronix zero-diffraction plate placed on a CaLCTec (Calabria Liquid Crystal Technology, Arcavacata di Rende, Italy) heating stage.

### 3.4. Electrochemical Studies

Electrochemical measurements were performed at ambient room temperature (∼20 °C) using GPES 4.9 software controlled by an Autolab PGSTAT 302potentiostat/galvanostat (EcoChemie, Utrecht, the Netherlands). A carbon nanotube (CNT) paste electrode with a disk diameter of 3.0 mm and a platinum plate characterized by a geometrical surface area of 1.0 cm^2^ were used as a working and counter electrode, respectively. A saturated calomel electrode (SCE) was used for a reference electrode. The saturated calomel reference electrode (SCE) was used to record the electrode potentials. The CNT paste working electrode was modified by simple film-casting method with **Zn/Fe**, **Zn** and ferrocene-containing precursor (**S4**) and named **Zn/Fe_CNT** paste, **Zn_CNT** paste and **Fc_CNT** paste electrodes, respectively. Their electrochemical behaviours were comparatively studied by cyclic voltammetry (CV) in 0.1 M NaOH supporting electrolyte. After immersion in the material, the electrode surface was dried at room temperature. For each sample, at least five scans of CV were recorded at a scan rate of 0.05 V∙s^−1^ to achieve the steady state of the modified electrode.

## 4. Conclusions

A new bimetallic coordination complex containing a pentacoordinated Zn(II) metal centre by a tridentate chelating terpyridine-based ligand and two monoanionic gallate ligands containing ferrocene units was synthesised and structurally characterized. The complex, like its previously reported structurally analogous parent complex (**Zn**) [3] lacking ferrocene units, exhibited liquid crystalline properties. While complex **Zn** arranged into a 3D columnar hexagonal mesophase, complex **Zn/Fe** organised into a regular 2D columnar hexagonal phase due to the presence of ferrocene units in the alkyl chain segments, preventing the segmentation of stacked molecules into groups.

Aiming to investigate the molecular relationship between structure and supramolecular properties and the relationship between structure and electrochemical properties, the electrochemical behaviour of complex **Zn/Fe** was characterized in comparison with the analogous **Zn** complex lacking ferrocene units and a ferrocene-containing organic derivative (precursor **S4**) by modification of a CNT paste electrode using a simple film-casting method. Different degrees of CNT paste electrode surface coverage were achieved depending on the modifier consistency and its affinity to CNT in the following order: **S4** > **Zn/Fe** > **Zn**.

The electrochemical features of **Zn/Fe_CNT** paste electrode consisted of a combination of **Zn_CNT** and **Fc_CNT** paste electrodes characterized by a good activity of the ferrocene/ferrocenium redox couple in the anodic branch. However, a quasireversible Fc/Fc+ couple in the **Zn/Fe_CNT** paste electrode vs. ideal reversible behaviour indicated a possible cooperative effect of the ferrocene within the zinc metallomesogen structure, assuring electron transfer to and from the redox centres of the ferrocene.

Comparison of the CVs of **Zn_CNT** paste electrode based on the 3D columnar hexagonal Zn(II) metallomesogen with the structural analogue organised in a smectic phase [24] revealed a possible contribution depending on the potential value of the cathodic peak characteristic of metallic zinc and zinc oxide formation. However, comparison of the electrochemical fingerprint of the liquid crystalline Zn(II) coordination complex organised in a columnar mesophase in the cathodic branch with that of the Fc/Fc+ in the anodic branch indicates the versatility of Zn/Fe metallomesogen for many electrochemical applications, e.g., sensing, catalysis, batteries, etc.

## Data Availability

Not applicable.

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
