# Peer review of "Hetero-Bimetallic Ferrocene-Containing Zinc(II)-Terpyridyl-Based Metallomesogen: Structural and Electrochemical Characterization"

_materials, 2023, doi:10.3390/ma16051946_

Round 1
Reviewer 1 Report
Dear author,
The following minor revisions are recommended in order to improve the manuscript.
1- "aspectroscopies" should be change to "spectroscopies" in page 3,line 88.
2- In page 3, line 106, the thermal behavior of the complex briefly reported. please add more details (suggestions) to interpret what happened in each weight loss steps.
Author Response
We are very grateful to the reviewer for his time and evaluation of our manuscript. We have addressed all the points raised, and we hope that our answers will be fully satisfying.
In reply we added the followings:
1- "aspectroscopies" should be change to "spectroscopies" in page 3, line 88.
Response: We apologize for the mistake, we changed in the text.
2- In page 3, line 106, the thermal behavior of the complex briefly reported. please add more details (suggestions) to interpret what happened in each weight loss steps.
Response: TGA analysis is used in this case to show the good thermal stability of the coordination complex, and the absence of unwanted decomposition within the mesophase temperature range (r.t. – 160oC), since the drawback of metallomesogens are usual low decomposition temperatures, before clearing into isotropic liquid. Moreover, TGA is used to indicate the presence of eventual hydration or crystallization solvent, that influenced in the structural analysis. We change the text in the manuscript according to the reviewer reccomendation:
“A loss of one molecule of water was observed at around 50oC, whilst the complex shows a good thermal stability [42]; the weight loss from degradation becomes significant above ca. 300°C and reaches 5% at T5% = 324°C (See Supplementary Materials: Figure S3). The majority of the organic part decompose between 300 – 500oC leaving a residual zinc and iron oxides of 8.17%.”

Reviewer 2 Report
1. The quality of these figures are poor, such as Fig. 3 and 4.
2. I think the current results can not support the exact coordination for the Zn(II) center, pls do the EA and MS.
3. Review all graphics, subtitles are small, ariel, no pattern. This seems irrelevant but it organizes the work for the reader.
4. “Some examples of binuclear metallomesogens found in literature are mainly reported as synthetic challenges with investigation of their supramolecular “soft” assemblies” Some related and updated refs could be cited, such as Micropor. Mesopor. Mat, 341(2022) 112098 and Inorganics, 10(2022) 202.
5. Table 3 has a better format, why haven't you established a standard?
6. How about the electrochemical characterization for EIS Nyquist?
Author Response
We are very grateful to the reviewer for his time and evaluation of our manuscript. We particularly appreciate the pertinence of his comments to improve our manuscript and to enhance the interest towards the broad scientific community reached by Materials. We have addressed all the points raised by the reviewer, and we hope that our answers will be fully satisfying.
In reply we added the followings:
Reviewer 2:
- The quality of these figures are poor, such as Fig. 3 and 4.
Response: We increased the qualities of the figures 3 and 4.
- I think the current results can not support the exact coordination for the Zn(II) center, pls do the EA and MS.
Response: The data regarding Elemental Analysis (EA) and Atomic absorption spectroscopy (AAS) are reported in the Supplementary Materials part:
“1. Experimental section. Synthesis.
Synthesis of Zn/Fe
Anal. Calcd. for C129H190Fe2N4O10Zn·1.0 H2O (2152.00 g·mol-1): C, 72.00; H, 8.99; N, 2.60. Found: C, 72.08; H, 9.12; N, 2.71%. AAS: Zn% calcd.: 3.15, found: 3.08.”
The structure is proposed based on similar coordination complexes reported in literature. We added references 38-41. Moreover, the integrals of the NMR aromatic proton peaks show a 2 to 1 benzoate to terpyridine ligand ratio, while the FT-IR spectra shows a monodentate anionic coordination of the functional benzoate ligands. Unfortunately, we do not have access to Mass Spectroscopy, however the results of EA and AAS support fully the structure and high purity of the coordination complex Zn/Fe.
- Review all graphics, subtitles are small, ariel, no pattern. This seems irrelevant but it organizes the work for the reader.
Response: We reviewed all graphics.
- “Some examples of binuclear metallomesogens found in literature are mainly reported as synthetic challenges with investigation of their supramolecular “soft” assemblies” Some related and updated refs could be cited, such as Micropor. Mesopor. Mat, 341(2022) 112098 and Inorganics, 10(2022) 202.
Response: We thank the referee for the suggestion, but, considering that the articles proposed does not seems to be strictly related to binuclear metallomesogens (they report MOF structures based on one type of metal) we added the second article as citation for the thermal stability [42].
- Table 3 has a better format, why haven't you established a standard?
Response: We redraw the tables according to the MDPI format.
- How about the electrochemical characterization for EIS Nyquist?
Response: We thank the reviewer for the suggestion, however EIS Nyquist plots give information about the double layer capacitance and charge transfer resistance, which depends on the surface coverage degree and do not provide more useful and clear electrochemical information related to the redox systems during potential scanning.

Round 2
Reviewer 2 Report
accepted.